# A Three-Phase Transport Model for High-Temperature Concrete Simulations Validated with X-ray CT Data

**DOI:** 10.3390/ma14175047

**Published:** 2021-09-03

**Authors:** Christoph Pohl, Vít Šmilauer, Jörg F. Unger

**Affiliations:** 1Federal Institute for Materials Research and Testing BAM, Unter den Eichen 87, 12205 Berlin, Germany; christoph.pohl@bam.de; 2Department of Mechanics, Faculty of Civil Engineering, Czech Technical University in Prague, Thákurova 7, 166 29 Prague 6, Czech Republic; vit.smilauer@fsv.cvut.cz

**Keywords:** concrete, porous media, spalling, dehydration, moisture transport, heat transfer, pore pressure, porosity, finite elements

## Abstract

Concrete exposure to high temperatures induces thermo-hygral phenomena, causing water phase changes, buildup of pore pressure and vulnerability to spalling. In order to predict these phenomena under various conditions, a three-phase transport model is proposed. The model is validated on X-ray CT data up to 320 °C, showing good agreement of the temperature profiles and moisture changes. A dehydration description, traditionally derived from thermogravimetric analysis, was replaced by a formulation based on data from neutron radiography. In addition, treating porosity and dehydration evolution as independent processes, previous approaches do not fulfil the solid mass balance. As a consequence, a new formulation is proposed that introduces the porosity as an independent variable, ensuring the latter condition.

## 1. Introduction

Exposing concrete structures to fire can lead to spalling, a process where material violently breaks off of the surface. How susceptible a concrete mixture is to spalling is therefore an important criterion for the safety and service life of concrete structures. The prevailing hypothesis is that spalling is caused by a combination of high pore pressures and thermal stresses. When concrete is exposed to high temperatures, dehydration and free water evaporation often exceed the rate of vapour migration, resulting in rising pore pressures. Due to its low permeability, this is especially severe for high performance concrete (HPC), making it generally more susceptible to spalling. With a numerical analysis of the thermo-hygral transport in porous media in general, and of concrete in particular, insights into the mechanisms and influencing factors can be gained.

Many different approaches have been proposed [1,2,3], where the most accurate are the three-phase models [4,5,6], i.e., models that consider liquid water, moist air and the solid matrix as separate phases. These are usually validated using temperature and pressure gauges at various positions within the specimen. The pressure measurements in particular can be problematic. The presence of the sensor influences the resulting cement matrix around it, and localized cracks are more likely to develop in its vicinity [7]. Furthermore, the size of the sensor is much larger than typical pore diameters, resulting in an averaged value over the measured area. To overcome these experimental difficulties, non-destructive computer tomography measurements are increasingly used. Using these CT results for validation of the thermo-hygral transport model is an important improvement pursued in this work.

Balázs et al. have used X-ray CT to measure the change in density of the concrete specimen after undergoing a heating cycle [8]. A reduction in density by 2% to 4% was found after heating to 500 °C, with good agreement between CT and conventional methods. The use of CT can resolve these changes in space. In a separate fire test, the reduction in density primarily occurred close to the surface that was subjected to the fire. Henry et al. have used X-ray microtomography to characterize the change in total pore space and its connectivity after heating to 600 °C for an hour [9]. The total pore space was found to be roughly 1.5 times larger, while the connectivity had increased by 3.5 times. The influence of air and water re-curing on the pore space, as well as the pore size distribution, was also investigated. In contrast to the experiments in [8,9], data from Powierza et al. [10] was obtained in-situ during the high temperature exposure. This makes it ideal for validating the water content in the pores to a numerical model and is, therefore, used here. The measured water content shows that the majority of previous models do not accurately reflect the water transport inside the concrete, as already pointed out in [7]. Additionally, knowing the water content allows for a deeper investigation of the influence of different dehydration formulations, several of which are compared in Section 2.1 and Section 3.2. We are unaware of any such comparison in the literature. The suitability of thermogravimetric analysis as the basis for dehydration models will be discussed. There is a mass balance equation associated with every phase of the porous medium. Previous approaches did not solve the skeleton mass balance equation, assuming the error to be negligible. The novelty in this work is that the skeleton mass balance is solved along with the other balance equations, avoiding this error and resulting in one fewer constitutive equation. This also allows for dehydration descriptions that depend on the pore state, e.g., liquid water saturation or pore pressure, a significant extension compared to previous models. These modifications allow solving the coupled problem more accurately, while introducing the possibility of more complex dehydration models in the future.

The material is modelled as an isotropic homogeneous continuum. As a macroscopic model, it does not resolve the aggregates as discrete particles in the finite element mesh. The focus lies on the heat and moisture transport phenomena; coupling to mechanical deformation and damage evolution are possible extensions in future work. In the model, the aggregates are assumed to be inert, both chemically and physically, i.e., they release no dehydration water. Thermogravimetric analysis showed that basalt, granite and sandstone lose up to 3.1% of mass when heated to 1000 °C [11], with similar results reported in [12]. Above 600 °C, carbonate aggregates (limestone and dolomite) decompose into CaO and CO_2_ [12], resulting in additional mass loss. Concretes with four different aggregates exposed up to 1000 °C showed similar temperature-dependent tensile and compressive strength decrease [12]. This indicates that cement paste presents the weakest link in concrete. However, different aggregates contribute to microcrack evolution due to the thermal expansion mismatch and this will impact permeability, sorption isotherms and diffusivity.

We have made a conscious effort to make the reproduction of this work as convenient as possible. All the necessary code and data are openly available, a container with all the necessary software is provided, and running all the required steps is fully automated. See Section 4 for details.

## 2. Model Description

### 2.1. Dehydration and Porosity Descriptions from the Literature

The importance of taking dehydration and changes in porosity into account when modelling the behaviour of concrete at high temperatures is widely recognized in the literature. We will present a number of dehydration and porosity descriptions.

Dehydration is the chemical process whereby chemically-bound water from the hardened cement paste is released into the pore space as temperatures increase. Heating above 105 °C releases non-evaporable water (hydration water), which occupies part of the gel pores or is chemically bound in chemical phases. How much water is released is usually determined experimentally by thermogravimetric analysis (TGA). Different aggregate types exhibit different temperature stability. When using TGA data to model dehydration, the assumption that mass loss equates to water release only holds for aggregate types that are chemically and physically stable at high temperatures, such as quartzite or basalt. Below 800 K, most aggregates are chemically and physically stable. Most models in the literature are calibrated using TGA measurement data.

For better comparison, all models have been brought to the same form,
(1)$$m_{\mathrm{dehyd}}\left(T\right)=c\nu \Gamma \left(T\right),$$
where *c* is the cement content (kg m^−3^), ν is the mass ratio of maximally released water to cement content, and Γ is the dehydration degree (Γ∈[0,1]). A common assumption is that water loss below 105 °C is water evaporation, with dehydration remaining at zero; the constant $T_{d}$ = 378.15 K will be used in these cases.

In the formulation proposed by Pesavento, $\nu =f_{s}m$, where $f_{s}$ is the stoichiometric factor and *m* is the ageing degree [6]. Only the dehydration degree is assumed to be a temperature-dependent sigmoid function. Gawin et al. use the same product formulation as Pesavento, but propose a cubic polynomial for the dehydration degree [13].

The formulation by Tenchev et al. is equivalent to using a temperature-dependent ν, i.e., $m_{\mathrm{dehyd}}\left(T\right)=c\nu \left(T\right)$. The proposed formulation is a step-wise linear function. Dwaikat and Kodur use a very similar expression [14].

The description by Dal Pont and Ehrlacher is one of the few approaches that takes the kinetics of the dehydration process into account [15]. The dehydration evolution is described by the differential equation
(2)$${\dot{m}}_{\mathrm{dehyd}}=-\frac{1}{\tau }\left(m_{\mathrm{dehyd}}-m_{\mathrm{dehyd}}^{\mathrm{eq}}\left(T\right)\right),$$
where τ is the characteristic time of mass loss and $m_{\mathrm{dehyd}}^{\mathrm{eq}}$ is the amount of water created at equilibrium. The value for the characteristic time is given in [15] as three hours.

All of the previous approaches are based on thermogravimetric analysis curves. TGA data are highly influenced by sample size, heating rate and vapour pressure [16]. The data used in these approaches were obtained on very small samples, low heating rates and in dry conditions. Furthermore, the results of TGA also include the CO_2_ released from carbonates.

These conditions do not necessarily apply to specimens tested for water migration and spalling risk under high temperatures. As a way of correcting for these discrepancies while keeping the function-of-temperature approach, Dauti et al. perform an inverse analysis on measured neutron radiography data [7]. The result is a much steeper curve (see Figure 1). This result, however, cannot be directly used in other simulations, since it reflects the sample size, heating rate and pore vapour state of their experiments. The given graph also does not extend beyond 450 °C.

Since the authors Dauti et al. have not provided an analytic description of their dehydration model, a logistic function was fitted to the given graph to obtain a mathematical expression as,
(3)$$\Gamma \left(T\right)=\frac{a}{1+e^{-k(T-T_{0})}}.$$

For the curve presented in their paper, the best fit is obtained with a = 0.8219, k = 0.0876 K^−1^ and $T_{0}$ = 578.1 K.

Porosity is the ratio of pore space to the overall volume, $n=(V_{w}+V_{g})/V$, where $V_{w}$ and $V_{g}$ are the volumes of the liquid and gas constituents in the averaging volume element *V*. In cementitious systems up to 105 °C, the porosity is composed mainly of capillary pores (diameters approximately 10 nm–10 μm) [17], physically adsorbed water covering surfaces of gel particles, entrapped and entrained air and internal pores within aggregates. The evaporable water comprises free water residing in the capillary pores plus physically adsorbed water. The state of water and detailed description of a colloid C-S-H model with densities can be found elsewhere [18].

Different authors provided temperature-dependent porosity evolution, briefly summarized below and in Figure 2. Dauti et al. describe the porosity as a linear function [7],
(4)$$n=0.05+1.4\times 10^{-4}T.$$

The porosity evolution according to Tenchev et al. is an empirical function [4],
(5)$$n=f_{n}\left(T\right)+n_{0},$$
noting that factors such as dehydration, chemical decomposition of aggregates, thermal strains and microcracking contribute to its increase. Their actual description assumes the porosity to stay constant below 100 °C, three times the initial porosity above 800 °C and a cubic polynomial in the range between *T* ∈ [100 °C, 800 °C]:
(6)$$n=n_{0}\left\{\begin{array}{cc} 1 & \mathrm{for}\;T<100\;{}^{{}^{\circ}}C,\\ aT^{3}+bT^{2}+cT+d & \mathrm{for}\;100\;{}^{{}^{\circ}}C\leq T\leq 800\;{}^{{}^{\circ}}C,\\ 3 & \mathrm{for}\;T>800\;{}^{{}^{\circ}}C. \end{array}\right.$$


The coefficients of the polynomial are chosen such that the porosity is $C^{1}$ continuous. Dal Pont and Ehrlacher [15] give the change in porosity simply as $n=n_{0}+0.72\times 10^{-3}m_{\mathrm{dehyd}}$. In the publications by Gawin et al., the porosity is described as a simple linear function,
(7)$$n=n_{0}+A_{n}\left(T-T_{0}\right),$$
with initial porosities given between 0.06 and 0.087, and the constant $A_{n}$ in the range 1.63 × 10^−4^ K^−1^ to 1.95 × 10^−4^ K^−1^. A much larger range of initial porosities (between 0.0512 and 0.13) are reported in later papers [19,20]. They also note that dehydration is only one factor driving the change in porosity, preferring the above empiric relation to a direct, dehydration-only description.

Dwaikat and Kodur obtain the change in porosity from the sum of volume fractions [14]
(8)$$n=n_{0}+V_{\mathrm{dehyd}},$$
where $V_{\mathrm{dehyd}}$ is the volume of the dehydrated water and assumed to be $V_{\mathrm{dehyd}}=\frac{m_{\mathrm{dehyd}}}{\rho_{w}}$. However, this is in conflict with the skeleton mass balance, as discussed further in Section 2.2.

### 2.2. Balance Equations

The general form of the mass balance equations, namely of dry air, water vapour, liquid water and the solid matrix, is
(9)$$\frac{\partial m_{\pi }}{\partial t}=r_{\pi }-\nabla J_{\pi },$$
where $m_{\pi }$ is the mass per unit volume, $J_{\pi }$ is the mass flux and $r_{\pi }$ is the mass source, each with respect to phase π={a,v,w,s}. The indices denote dry air, water vapour, liquid water and skeleton, respectively.

For the solid phase, the mass balance is given by
(10)$$\frac{\partial m_{s}}{\partial t}={\dot{m}}_{\mathrm{dehydr}}-\nabla \left(m_{s}v_{s}\right)$$
with the skeleton mass per unit volume $m_{s}=\left(1-n\right)\rho_{s}$. This equation is often not solved in other models [4,13]. Dehydration generally causes an increase in porosity and further chemical changes in the solid. The idea of a constant skeletal density has been used as a C-S-H bulk density description in the whole isotherm range [18]. In such a case, the C-S-H sheet maintains a constant skeletal density of 2850 kg m^−3^, and packing of the globules provides the correct C-S-H bulk density for different water contents [21]. Hydrated cement paste is mainly composed of C-S-H, together with portlandite, ettringite and other minor phases. We may extend the idea of a constant C-S-H skeletal density to the whole cement paste and concrete, resulting in a constant skeletal density $\rho_{s}$. Such a description simplifies the effect of dehydration. Note that concrete bulk density provides no information on the skeletal density. This means that only two of the quantities in the set $\left\{n,\rho_{s},{\dot{m}}_{\mathrm{dehydr}}\right\}$ are independent. For the cases discussed here, the mechanical deformation, and therefore $v_{s}$, is kept zero.

The dry air mass conservation is described by
(11)$$\frac{\partial m_{a}}{\partial t}=-\nabla J_{a},$$
where $m_{a}$ is the dry air mass per volume of a porous medium, and $J_{a}$ is the dry air mass flux. The mass per unit volume of dry air is given by
(12)$$m_{a}=nS_{g}\rho_{a}=n\left(1-S_{w}\right)\rho_{a},$$
where *n* is the porosity, $S_{g}$ and $S_{w}$ the gas and liquid water saturation, respectively, and $\rho_{a}$ is the density of dry air.

Free water is present in two different phases—as water vapour and as liquid water. This results in two mass balance equations,
(13)$$\begin{array}{cc} \frac{\partial m_{v}}{\partial t} & ={\dot{m}}_{\mathrm{vap}}-\nabla J_{v}, \end{array}$$
(14)$$\begin{array}{cc} \frac{\partial m_{w}}{\partial t} & =-{\dot{m}}_{\mathrm{vap}}-{\dot{m}}_{\mathrm{dehydr}}-\nabla J_{w}. \end{array}$$

These are commonly summed up to remove the evaporation term ${\dot{m}}_{\mathrm{vap}}$, resulting in a balance of total water,
(15)$$\frac{\partial (m_{v}+m_{w})}{\partial t}=-{\dot{m}}_{\mathrm{dehydr}}-\nabla \left(J_{v}+J_{w}\right).$$

The enthalpy balance can be written as
(16)$$\begin{array}{c} {\left(\rho C_{p}\right)}_{\mathrm{eff}}\frac{\partial T}{\partial t}+\left(\rho_{w}C_{pw}v_{w}+\rho_{g}C_{pg}v_{g}\right)\nabla T-\nabla \left(\lambda_{\mathrm{eff}}\nabla T\right)=\\ -{\dot{m}}_{\mathrm{vap}}\Delta H_{\mathrm{vap}}-\Delta H_{\mathrm{dehydr}}, \end{array}$$
where ${\left(\rho C_{p}\right)}_{\mathrm{eff}}$ is the effective heat capacity of the concrete, including water and moist air contributions, $C_{pw}$ is the specific heat capacity of liquid water, $C_{pg}$ is the specific heat capacity of moist air and $\lambda_{\mathrm{eff}}$ the effective thermal conductivity of the concrete.

The vapour mass source is
(17)$${\dot{m}}_{\mathrm{vap}}=-\frac{d}{dt}\left(\rho_{w}nS_{w}\right)-\nabla \left(nS_{w}\rho_{w}v_{w}\right).$$

### 2.3. Constitutive Equations

The dry air and water vapour flows are composed of an advective and a diffusive part,
(18)$$\begin{array}{cc} J_{a} & =n\left(1-S_{w}\right)\rho_{a}v_{g}+n\left(1-S_{w}\right)\rho_{g}D_{AV}\nabla \left(\frac{\rho_{a}}{\rho_{g}}\right), \end{array}$$
(19)$$\begin{array}{cc} J_{v} & =n\left(1-S_{w}\right)\rho_{v}v_{g}+n\left(1-S_{w}\right)\rho_{g}D_{VA}\nabla \left(\frac{\rho_{v}}{\rho_{g}}\right), \end{array}$$
where $v_{g}$ is the gas velocity. For binary mixtures such as moist air, the rates of mass diffusion are of equal magnitude and opposite direction for constant total concentration. Therefore the gas diffusion coefficient is $D=D_{AV}=D_{VA}$. Additionally, it is assumed to be constant. The liquid water flow is only advective,
(20)$$J_{w}=nS_{w}\rho_{w}v_{w},$$
where $v_{w}$ is the liquid water velocity.

The advection is described by Darcy’s law,
(21)$$\begin{array}{cc} n\left(1-S_{w}\right)v_{g} & =-\frac{kk_{rg}}{\mu_{g}}\nabla p_{g}, \end{array}$$
(22)$$\begin{array}{cc} nS_{w}v_{w} & =-\frac{kk_{rw}}{\mu_{w}}\nabla p_{w}. \end{array}$$

The intrinsic permeability *k* increases as the temperature rises. This is due to both increasing porosity caused by dehydration and crack creation as a result of thermomechanical stresses. Here, it is given by
(23)$$k=k_{0}10^{A_{k}\left(T-T_{\mathrm{ref}}\right)},$$
where $k_{0}$ is the intrinsic permeability at the reference temperature $T_{\mathrm{ref}}$ and $A_{k}$ is a material constant. This is a modification of the formulation proposed in [19], which specifies further terms for capturing gas pressure and damage dependence. The relative permeabilities depend on the current saturation with several models having been proposed. Here, we are using a description given by Beneš and Štefan [22],
(24)$$\begin{array}{cc} k_{rg} & =10^{S_{w}\psi }-S_{w}10^{\psi }, \end{array}$$
(25)$$\begin{array}{cc} k_{rw} & =10^{(1-S_{w})\psi }-\left(1-S_{w}\right)10^{\psi }, \end{array}$$
with ψ=0.05−22.5n.

A common choice for the sorption isotherms, also adopted here, is the model proposed by Baroghel-Bouny et al. [23]. Based on fitting experimental data for four different concrete mixes, the following formulation is obtained:
(26)$$p_{c}\left(S_{w}\right)=a{\left(S_{w}^{-b}-1\right)}^{1-1/b},$$
where *a* and *b* are material parameters. The hysteresis between sorption and desorption branches is neglected. Solving for the saturation gives
(27)$$S_{w}={\left({\left(\frac{p_{c}}{a}\right)}^{\frac{b}{b-1}}+1\right)}^{\frac{-1}{b}}.$$

The capillary pressure $p_{c}$ can be calculated from the Kelvin equation,
(28)$$p_{c}=\rho_{w}\frac{RT}{M_{w}}\ln\left(\frac{p_{v}}{p_{vs}}\right),$$
where $p_{vs}$ is the saturation vapour pressure.

The effective heat capacity ${\left(\rho C_{p}\right)}_{\mathrm{eff}}$ is taken as a weighted sum of the specific heat capacities of the individual components,
(29)$${\left(\rho C_{p}\right)}_{\mathrm{eff}}=m_{s}C_{p}^{s}+m_{w}C_{p}^{w}+m_{v}C_{p}^{v}+m_{a}C_{p}^{a}.$$

The specific heat capacity of the skeleton according to [24] is
(30)$$C_{p}^{s}=C_{p0}^{s}+a\left(T-T_{\mathrm{ref}}\right)-b{\left(\frac{T-T_{\mathrm{ref}}}{120\;K}\right)}^{2},$$
where a = 0.666 J K^−2^, b = 4 J K^−1^ and $T_{\mathrm{ref}}$ = 295 K. The heat capacities of liquid water, water vapour and dry air can be found in Appendix A. The dehydration enthalpy $\Delta H_{\mathrm{dehydr}}$ is assumed to be constant [22].

The effective thermal conductivity $\lambda_{\mathrm{eff}}$ is the product of the dry thermal conductivity $\lambda_{\mathrm{dry}}$ and a factor for the increased thermal conductivity of pore water [5]
(31)$$\lambda_{\mathrm{eff}}=\lambda_{\mathrm{dry}}\left(1+\frac{4n\rho_{w}S_{w}}{\left(1-n\right)\rho_{s}}\right),$$
where the dry thermal conductivity depends on the temperature
(32)$$\lambda_{\mathrm{dry}}=\lambda_{\mathrm{dry}}^{0}\left(1+A_{\lambda }\left(T-T_{\mathrm{ref}}\right)\right).$$

### 2.4. Boundary Conditions

The temperature boundary conditions are either Dirichlet conditions
(33)$$T=\hat{T}\;\mathrm{on}\;\Gamma_{T}^{1}$$
or convective-radiative Robin conditions
(34)$$q_{T}=h\left(T-T_{\infty }\right)+\varepsilon \sigma \left(T^{4}-T_{\infty }^{4}\right)\;\mathrm{on}\;\Gamma_{T}^{2},$$
where *h*, ε and σ are the convective heat transfer coefficient, the surface emissivity and the Stefan-Boltzmann constant, respectively. The air pressure is prescribed via a Dirichlet condition
(35)$$p_{a}=p_{a\infty }\;\mathrm{on}\;\Gamma .$$

Lastly, the boundary condition for the vapour transport is also a Robin condition,
(36)$$q_{v}=\beta_{c}\left(\rho_{v}-\rho_{v\infty }\right)\;\mathrm{on}\;\Gamma ,$$
where $\beta_{c}$ is the convective water vapour transfer coefficient.

### 2.5. Numerical Approximation

The set of balance Equations (Equation 10), (Equation 11), (Equation 15) and (Equation 16) is solved via the finite element method. When solving only the air mass balance, water mass balance and enthalpy balance, the set of independent variables is $\left\{p_{v},p_{a},T\right\}$. In the case where, additionally, the skeleton mass balance is solved for, the porosity *n* enters as a fourth independent variable. The choice of independent variable is not arbitrary. Gawin et al. argue that the capillary pressure $p_{c}$ can be used, even beyond the critical point of water, by reinterpreting the resulting values as a substitute for the product of water potential Ψ and liquid water density $\rho_{w}$ [19]. In our experience, the choice of $\left\{p_{g},p_{c},T\right\}$ leads to poorer convergence.

Solving the additional skeleton mass balance results in a thermodynamically consistent solution, and in addition allows for one fewer constitutive equation. The characterization of the material may then be easier and faster, or measurements of the corresponding material parameter—porosity in this case—can be used as additional verification.

The domain is discretized with mixed elements, with a Lagrange element for each component. The vector of degrees of freedom of the unknowns can be written as
(37)$$u=\left[\begin{array}{c} p_{v}\\ p_{a}\\ T\\ n \end{array}\right].$$

The polynomial order is the same for all subelements, and linear elements were found to suffice. Increasing the order or choosing different orders for the individual components is trivial due to the implementation in FEniCS [25]. Since the test function *w* is from the same function space, it can also be split into its components $w_{\pi }$.

To discretize Equations (Equation 10), (Equation 11), (Equation 15) and (Equation 16), they are transformed into their weak forms. The residual for the dry air mass balance is
(38)$$r_{a}=\int_{\Omega }w_{a}\frac{\partial m_{a}}{\partial t}dx-\int_{\Omega }\nabla w_{a}J_{a}dx,$$
where $J_{a}$ is the dry air flux from Equation (Equation 18). The water mass balance reads
(39)$$r_{w}=\int_{\Omega }w_{w}\frac{\partial (m_{v}+m_{w})}{\partial t}dx-\int_{\Omega }\nabla w_{w}\left(J_{v}+J_{w}\right)dx+\int_{\Omega }w_{w}{\dot{m}}_{\mathrm{dehydr}}dx,$$
with the vapour flux $J_{v}$ and the liquid water flux given in Equations (19) and (20), respectively. The skeleton mass balance residual is
(40)$$r_{s}=\int_{\Omega }w_{s}\frac{\partial m_{s}}{\partial t}dx-\int_{\Omega }w_{s}{\dot{m}}_{\mathrm{dehydr}}dx.$$

Lastly, the residual for the enthalpy balance,
(41)$$\begin{array}{cc} r_{T}= & \int_{\Omega }w_{T}\;{\left(\rho C_{p}\right)}_{\mathrm{eff}}\;\frac{\partial T}{\partial t}dx+\int_{\Omega }w_{T}\left(\rho_{w}C_{pw}v_{w}+\rho_{g}C_{pg}v_{g}\right)\nabla Tdx+\int_{\Omega }\nabla w_{T}\left(\lambda_{\mathrm{eff}}\nabla T\right)dx\\ + & \int_{\Omega }w_{T}\;{\dot{m}}_{\mathrm{vap}}\Delta H_{\mathrm{vap}}dx+\int_{\Omega }w_{T}{\dot{m}}_{\mathrm{dehydr}}\;\Delta H_{\mathrm{dehydr}}dx. \end{array}$$

Since the dehydration is assumed to be an irreversible process, the dehydration degree is an internal variable that only depends on the maximum temperature at each material point,
(42)$$\Gamma \left(t\right)=\Gamma (T_{\max}\left(t\right)).$$

Discretization in time is achieved by applying the Rothe method, resulting in a nonlinear set of equations
(43)$$r\left(u^{n+1},u^{n}\right)=\left(\begin{array}{c} r_{w}\left(u^{n+1},u^{n}\right)\\ r_{a}\left(u^{n+1},u^{n}\right)\\ r_{T}\left(u^{n+1},u^{n}\right)\\ r_{s}\left(u^{n+1},u^{n}\right) \end{array}\right)=0.$$

The time integration is performed using an Euler backward method with adaptive time stepping.

## 3. Results

### 3.1. Skeleton Mass Density

Porosity and dehydration evolution are dependent processes, see Equation (Equation 10). Their evolution influences air, water and heat transfer, and their interdependence should not be neglected as in the previous models under discussion [4,6,7,13,14,15,24]. As a result, the skeleton mass density is often treated inconsistently. It could be argued that the skeleton mass density changes in such a way as to fulfil the mass balance for two independently chosen descriptions of the porosity and dehydration. However, all models assume it to be constant when it enters into other equations, such as the formulation for the heat capacity.

Given the skeleton mass balance and a pair of dehydration and porosity models, one can solve the balance equation for a material point to obtain the skeleton mass density. To do so, the skeleton mass balance $\frac{\partial m_{s}}{\partial t}={\dot{m}}_{\mathrm{dehydr}}$ can be rearranged into an ODE for the skeleton mass density,
(44)$$\frac{d\rho_{s}}{dT}=\frac{1}{1-n}\left(\frac{dm_{\mathrm{dehydr}}}{dT}+\rho_{s}\frac{dn}{dT}\right).$$

This gives the theoretical evolution of that density such that the skeleton mass balance is maintained and can be seen for all the models under discussion in Figure 3.

The resulting values are higher than one would expect. Dry cement mixed with dry quartzitic aggregates in the same ratio as the concrete in Section 3.2 would result in a density of about 2740 kg m^−3^. By hydration and curing, and then further dehydration by heating, no increase in skeletal density is possible. This demonstrates the problematic consequences of independently choosing porosity and dehydration descriptions.

Solving the skeleton mass balance directly links the dehydration and porosity evolution in our case. We assume the skeletal density to be constant. The additional balance equation allows for one fewer constitutive equation, thereby either saving on experimental effort or providing additional data for validation. If the dehydration description solely depends on temperature, the balance equation becomes an ODE that could be solved separately. Discretizing the skeleton mass balance along with the other balance equations allows for more complex dehydration descriptions, depending not only on temperature but also on, for example, liquid water content or pore pressure.

### 3.2. Validation on a Slowly Heated Cylinder

The data for validation have been obtained from an X-ray computer tomography (CT) scan of a concrete sample at various times during heating, published in [10]. A cylindrical specimen (⌀40 × 100 mm) of high-strength concrete was surrounded by a glass-ceramic shell and then wrapped in aluminium-silicate wool to hinder the exchange of moisture and heat along the lateral boundary. The mix of the concrete can be seen in Table 1. The concrete has a compressive strength of 104 MPa [26]. The specimens were stored under water for 28 days after casting. At the time of measurement, they were at least 90 days old, having been stored in a climate chamber of 20 °C and 65% relative humidity between water storage and heating. Heating was applied using an electric heating element at the top surface, with a heating rate of 10 K min^−1^ for the first 28 min up to a maximum temperature of 320 °C. This temperature was then kept constant for another 130 min.

The specimen was placed on a spinning table to take multiple images from different angles, and then reconstruct a 3D volume representation from the projections. Taking a single image took roughly one second, and a full scan was comprised of 650 projections, resulting in about 10 min per scan. As a consequence, the values at the given times are averaged values of 10 min around this point in time. The CT scans result in grayscale images where the luminance corresponds to the density of the material. The difference in brightness between empty and fully saturated pores was taken to represent 100% moisture change. The moisture change can thus be quantified by the change in brightness between the initial image before heating and subsequent images during heating. Deformation was corrected for using digital volume correlation. Since embedded temperature sensors would introduce artefacts into the X-ray images, separate experiments with embedded thermocouples were performed to find the temperature distribution. The top 5 mm of the images were removed, because they exhibit cone-beam artefacts rather than actual material behaviour. Further details of the experimental setup and image correction can be found in [10]. The images from the CT clearly show the change in moisture as the drying front advances into the material, see Figure 4. Averaging over the width of the diameter gives a one-dimensional moisture distribution, which we will use for comparison to the finite element model.

The initial conditions are given as a constant vapour pressure corresponding to a relative humidity of 65%, a dry air pressure of ambient air pressure (101.325 kPa) minus the vapour pressure, a temperature of 295 K and an initial porosity of 7.2455% as in [10]. Inhomogeneous initial moisture conditions due to drying during storage were initially considered, but found to be of little influence. The material parameters used in the simulation can be seen in Table 2.

Additionally, the temperature has been measured with four temperature gauges along the axial direction. A comparison of these results with the simulation can be seen in Figure 5.

For comparison to the CT data, the change in moisture content is computed by
(45)$$\Delta m_{m}=m_{w}-m_{w}^{0}+m_{v}-m_{v}^{0}+m_{\mathrm{dehydr}}^{\max}\left(\Gamma -1\right),$$
where $m_{\mathrm{dehydr}}^{\max}$ is the maximally dehydrated water mass when the dehydration degree Γ reaches one. The model shows good agreement, as illustrated in Figure 6. Particularly for the early part (15 and 30 min) and towards the end of the experiment (135 min), the location of the drying front of the simulation coincides with the experimental results. For the times in between (45, 60, 90 min), the model predicts a slightly more advanced front than can be observed in the experiments. The moisture accumulation behind the drying front, i.e., the moisture clog, is slightly overestimated at 30 min, and slightly underestimated for later times. The steep gradient of the moisture content is captured well. In the colder interior of the specimen, an increase in moisture content can be seen in the CT data. This broader accumulation of moisture is not present in the model results. There are two possible causes for this. Firstly, the dehydration description based on Equation (Equation 3) releases basically no water below 200 °C (the temperature range of this region), which leads to an underestimation of the available water. Secondly, the analysis and postprocessing to get from CT brightness changes to water mass loss may introduce a systematic error as the experiment goes on.

The simulation allows for further insight into the state of water inside the pores, as well as the evolution of the solid phase. In Figure 7, the saturation reaches almost zero on the exposed surface, with a steep gradient, especially for the first 45 min. The moisture clog shows up again here. The maxima in saturation and gas pressure occur at the same time, yet not at the same locations. The maxima of the gas pressure are in front of the highest water accumulation, with values exceeding 30 MPa. For mechanical loads on the skeleton, the pore pressure is the relevant quantity, which is a purely postprocessed quantity given by
(46)$$p_{\mathrm{pore}}=S_{w}p_{w}+\left(1-S_{w}\right)p_{g}-p_{g,\infty }.$$

Due to the slow heating and comparatively low temperatures, the maximum pore pressure is about 5 MPa. The water pressure shows very large negative values in the dry regions of the specimen, but due to the low saturation there, the resulting influence on the pore pressure remains in the order of that of the gas pressure. As a result, the pore pressure remains mostly negative, since the capillary effect of the liquid water is larger than the increase in gas pressure. The porosity exhibits an almost step-like evolution due to the similarly steep dehydration formulation. After 60 min, it no longer changes since the temperature does not rise any further.

Of particular interest for this work is the choice of dehydration formulation. As mentioned in Section 2.1, dehydration conditions for TGA will differ significantly from those in a specimen for spalling tests. The models have all been recalibrated to the TGA data for this particular concrete. This was achieved using a nonlinear least squares method. The result of the calibration can be seen in Figure 8, and the parameters are shown in Appendix B. Note that while the concrete under investigation has quartzitic aggregates, the comparatively low temperatures of the experiment allow for experiment and modelling to be repeated for other aggregate types without difficulties.

The simulation was repeated for each of the dehydration models that have been presented. The change in moisture mass shows significant differences between the models, see Figure 9. In particular, none of the models based on TGA exhibit enough dehydration during the early heating phase to match the experimental results. The dehydration description from Tenchev et al. leads to a loss of convergence beyond 21 min. The values during the first 21 min of heating are shown for completeness.

### 3.3. High-Temperature Benchmark Problem

While the CT data from the previous example enable direct validation of the moisture content inside the specimen, it is limited to temperatures of about 320 °C. To show that the proposed model also works beyond the critical point of water, and to allow comparison to other models, a common benchmark problem is presented here.

It was proposed by Tenchev et al. [4] and also solved in [24]. A cross-section of a concrete column exposed to fire is simulated. Symmetry in the *y*-direction makes this a one-dimensional problem. The surface of the column is exposed to fire according to the ISO-834 fire curve
(47)$$T=293.15\;K+345\;K\log\left(\frac{2t}{15\;s}+1\right)$$
for one hour. The outside atmosphere has a pressure of 0.1 MPa, and a relative humidity of 80%. The initial conditions for the concrete are a temperature of 20 °C, a gas pressure of also 0.1 MPa and vapour pressure equal to the saturation pressure, i.e., a relative humidity of 100%.

The resulting temperature and gas pressure distributions after 10, 30, and 60 min can be seen in Figure 10. For the temperature distribution, a very close match is obtained. Temperature information at 10 min. is not in the original paper. The gas pressure distribution appears to have a steeper decrease after the peak for our simulations.

In the paper that proposed this benchmark [4], an equidistant timestepping scheme with a timestep size of 2 s was used. The subsequent paper by Davie et al. [24] reduced it to 0.5 s. Both provided little justification for the specific choice of step size. Due to the highly nonlinear nature of the fire curve, an adaptive timestepping scheme can potentially save a significant amount of computing time. The criterion for increasing the time step is the number of Newton–Raphson iterations. If there are fewer than four iterations needed to solve the set of nonlinear equations, the new time step is set to 1.5 times the old time step. Consequently, much larger timesteps are used. Even with a conservative starting timestep of 5 s, the whole hour is integrated in just 87 timesteps, compared to 1800 or 7200 steps, respectively. The mean timestep was 41.44 s, and the largest timestep was 85.43 s. To compute the error introduced by larger timesteps, a reference simulation with equidistant timesteps of 0.01 s was performed. Equidistant timestepping with a timestep of two seconds as in [4] results in 0.0358, 0.0642 and 0.00202 as the $L_{2}$ norm of the relative errors for $p_{v}$, $p_{a}$ and *T*. In comparison, the adaptive scheme with the much larger timesteps also leads to larger errors, with 0.0506, 0.0927 and 0.00306, again for $p_{v}$, $p_{a}$ and *T*. For the largest error, that of $p_{a}$, using the adaptive scheme leads to an increase in the error of about three percentage points, while computational effort is reduced by a factor of about 20.

Solving the skeleton mass balance leads to a consistency between the porosity and dehydration evolutions. If they are chosen independently, as was the case with all the previous approaches, the results violate the skeleton mass balance. The change in skeleton mass consists of two terms, the change due to porosity change $\rho_{s}\left(n_{0}-n\right)$, and the change due to dehydration $m_{\mathrm{dehyd}}$. The incompatibility between the two terms can be seen in Figure 11.

The example shows that the model is able to cope with the transition beyond the critical point of water. Using an adaptive timestepping scheme can drastically reduce computation time. The introduction of the skeleton mass balance into the set of equations to be solved avoids incompatibility between dehydration and porosity descriptions.

## 4. Replication

This work, including all the tables and figures, is intended to be easily and fully repeatable. Reproducibility and repeatability are important parts of cumulative science. For a general introduction to reproducible computational research, see [27]. To allow repetition, finding mistakes or discrepancies between text and source code, as well as extend the analysis to new models or new data, an approach that is straightforward to use has been implemented in this work. The following section will explain its components.

Firstly, all of the input data, source code, parameters and scripts should be publicly available. In our case, all of the data are available at Zenodo, via the DOI 10.5281/zenodo.4452800 (accessed on 27 August 2021). This includes all the simulation code, TGA data, plotting instructions and LATEX files.

Secondly, in order to reproduce the results, not only is it necessary to have the source code and inputs but also the environment under which the code was executed, that is, the exact versions of all the programs and libraries used. We have chosen to provide a reproducible environment (or at least a reasonable approximation) in the form of a Docker container with all the required dependencies installed. Not only does this allow anyone to rerun the code in the same computational environment but also makes it very convenient. There is no need to manually install all the software that was used, which can be cumbersome and difficult. In addition, the Dockerfile serves as a readable description of all dependencies and the environmental setup. The container is available on Dockerhub at christophpohl/paper.

Lastly, and most importantly, all the data processing should be fully automated. This way, inefficient and error-prone manual steps are avoided, and a clear chain of reproducible steps is obtained. From running the simulation, via creating the graphs, to compiling the final PDF, all programs are invoked using a build automation tool. If reproducibility is the only criterion, this is not strictly necessary. A slightly simpler shell script that runs all required steps in sequence would suffice. However, changes in any of the files would require that everything is run again from scratch, even if only changes in the graphs or wording of small sections were made.

The use of a build automation tool avoids this by encoding the dependencies between the steps, their dependencies and their outputs as a directed acyclic graph (DAG). A visual representation of such a graph can be seen in Figure 12. The build tool will then assemble the DAG, and when asked to build the final PDF, traverse it to find the tasks that are not up to date and run them. A task is not up to date when either its output is missing or one of its dependencies has changed since the last time it was run. That way, changes in any of the predecessors will always trigger a reevaluation of all necessary tasks, and no running of unnecessary steps is taking place.

## 5. Conclusions and Outlook

Dehydration and its influence on the solid skeleton are key processes in understanding the moisture transport in concrete at high temperatures. Data from thermogravimetric analysis, which is performed on ground up concrete in artificial atmospheres, are not suitable for deriving a constitutive equation of dehydration for compact specimens under high heating rates. Comparing the predictions of existing models to X-ray CT data reveals drastic discrepancies, mainly related to their choice of dehydration description. Additionally, an inconsistency is introduced in these models by treating dehydration and porosity evolution independently. Solving the additional balance equation of skeleton mass yields the porosity as a result of the dehydration. That way, there is one less constitutive equation, and experimental data on the porosity may be used for additional validation.

The dehydration description proposed by Dauti et al. which is used here is based on an inverse analysis [7], and as a result is only valid within the bounds of the experimental setup. Extension to higher temperatures requires further experiments. Driving the research in this field is the need for higher spalling resistance, particularly for high performance concrete. Therefore, coupling the preceding model to damage mechanics in the search for better spalling prediction is a natural next step. Furthermore, mesoscale modelling of this problem might reveal interesting effects, as the interfacial transition zone has a higher porosity and connectivity [28], whereas the aggregates act as a barrier to moisture migration, increasing the tortuosity of the transport path. Lastly, obtaining a set of parameters for the constitutive equations is laborious, even for just one concrete. A publicly accessible repository of material data for concrete would greatly benefit the research, not only in this field.

## Figures and Tables

**Figure 1 materials-14-05047-f001:**
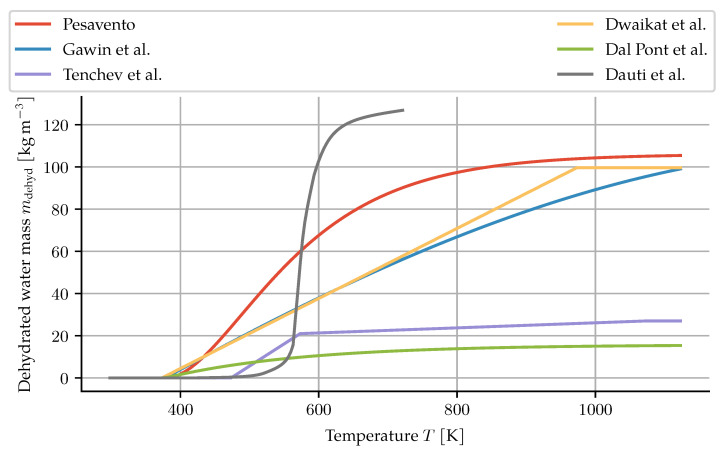
Comparison of dehydration descriptions. Parameter values as given in the respective papers. For the description by Dal Pont and Ehrlacher, the equilibrium dehydration mass is plotted.

**Figure 2 materials-14-05047-f002:**
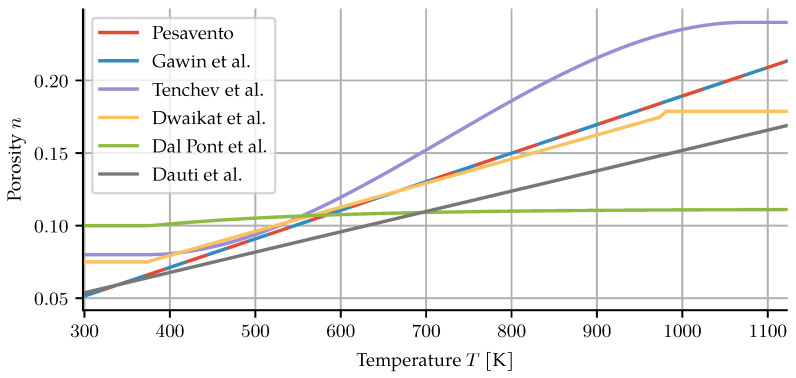
Comparison of porosity descriptions.

**Figure 3 materials-14-05047-f003:**
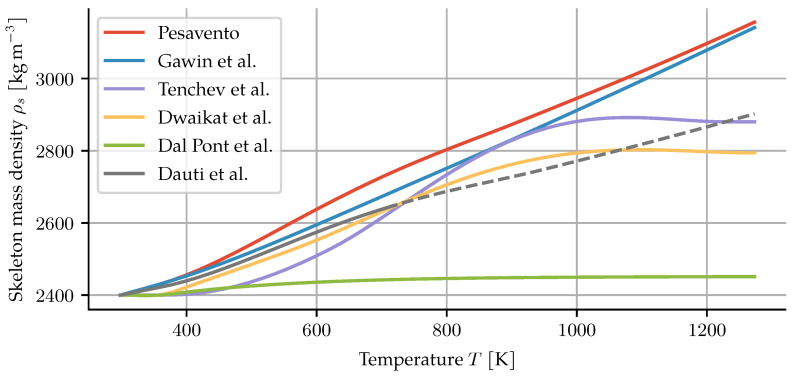
Theoretical skeleton mass density for given combinations of dehydration and porosity models.

**Figure 4 materials-14-05047-f004:**
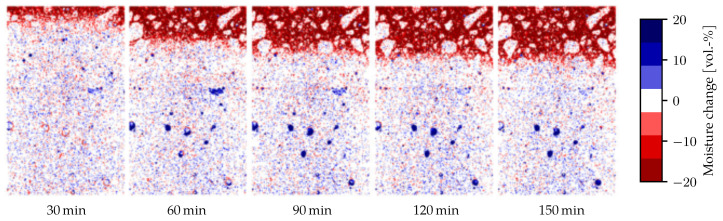
Change is moisture during the heating process. An advancing drying front (in red), as well as the filling of macropores (in blue) in the lower part of the specimen, can be seen. Image from [10].

**Figure 5 materials-14-05047-f005:**
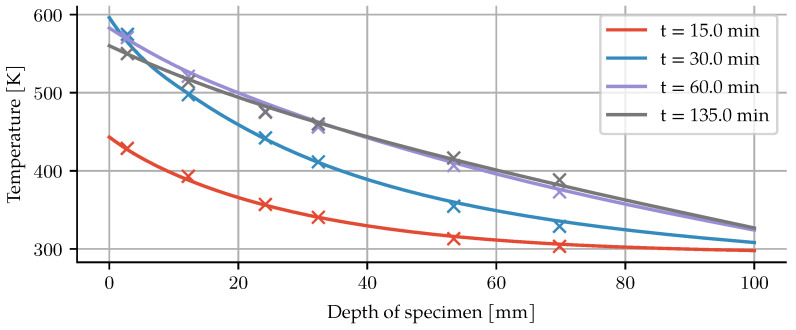
Comparison of temperature evolution between experiment (crosses) and numerical model (lines).

**Figure 6 materials-14-05047-f006:**
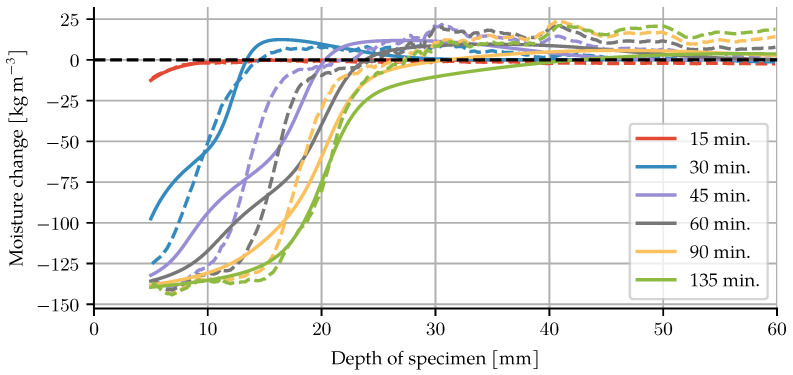
Comparison of water loss between CT data (dashed lines) and numerical model (solid lines).

**Figure 7 materials-14-05047-f007:**
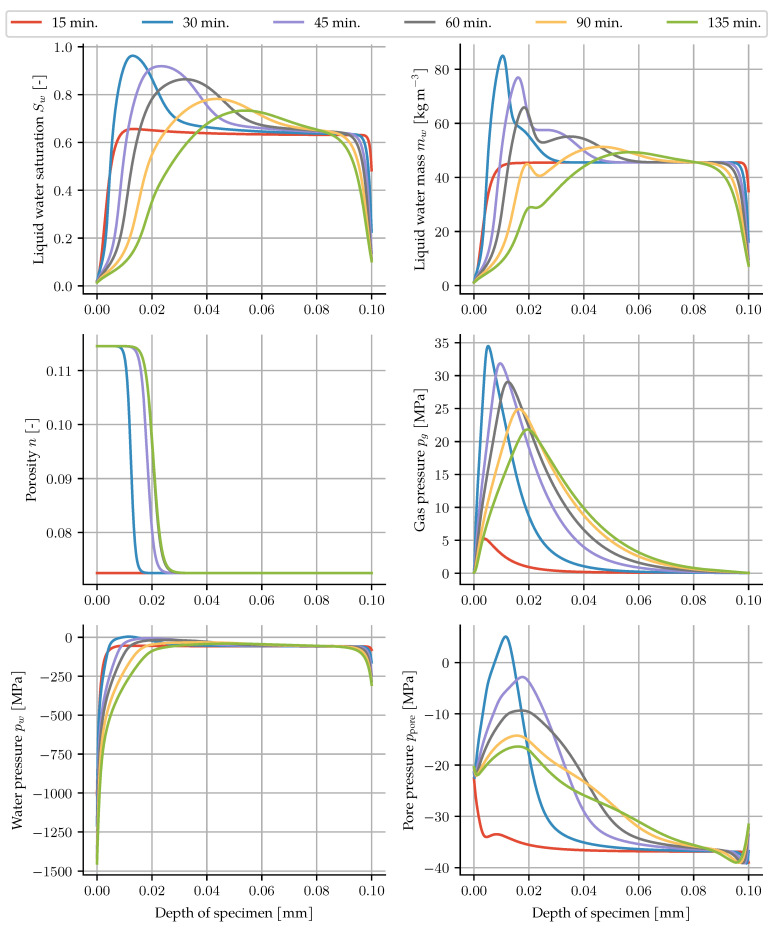
Resulting saturation, gas pressure, liquid water mass, porosity, gas pressure, water pressure and pore pressure for each of the measurement times.

**Figure 8 materials-14-05047-f008:**
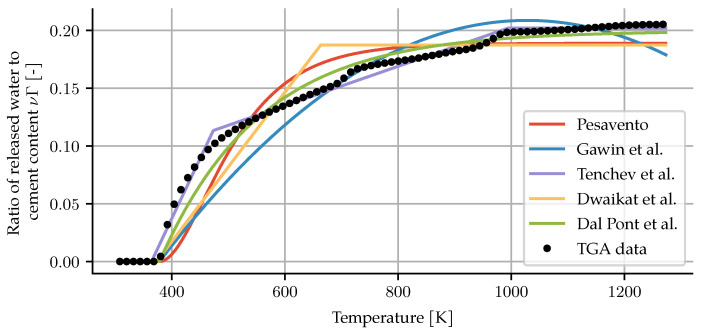
Calibration of TGA-based models for the concrete of the specimen.

**Figure 9 materials-14-05047-f009:**
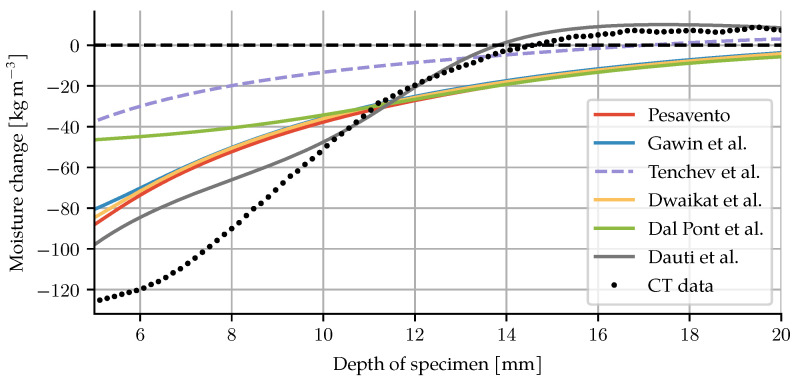
Change in moisture mass after 30 min for each of the presented dehydration models. The black dots represent the CT data. The combination of our model with the Tenchev dehydration did not converge to a solution beyond 21 min. The dashed line shows the results at that time.

**Figure 10 materials-14-05047-f010:**
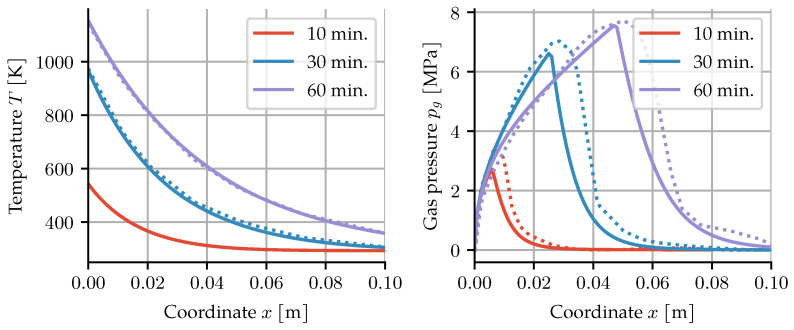
Temperature and gas pressure distribution along the column section. The dotted lines show the results by Davie et al. [24], whereas the solid lines are our results.

**Figure 11 materials-14-05047-f011:**
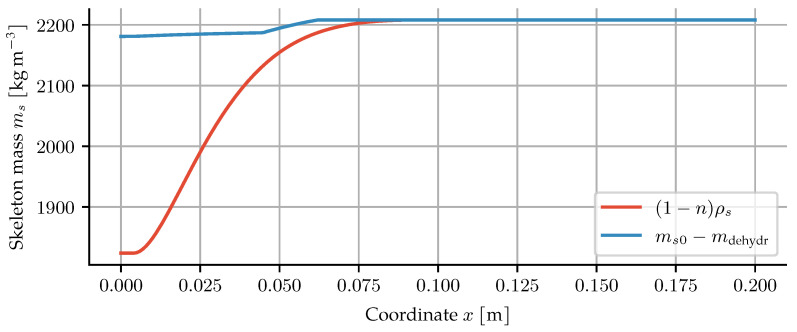
Incompatible skeleton mass after 60 min of heating for the model based on [24].

**Figure 12 materials-14-05047-f012:**
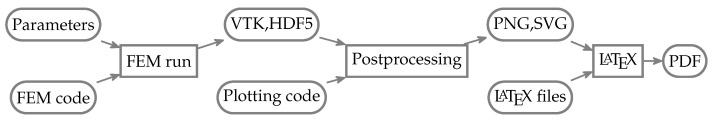
Dependency graph of prototypical numerical methods paper.

**Table 1 materials-14-05047-t001:** Mixture of the high-strength concrete [10].

Component	Content (kg m^−3^)
Cement CEM I 42.5 R	580
Water	173
Quarzitic aggregate	
0/2 mm	764
2/4 mm	229
4/8 mm	535
Silica fume	63.8
Superplasticizer	14.5

**Table 2 materials-14-05047-t002:** Material parameters used in the simulation.

Parameter	Value	Unit	Parameter	Value	Unit
$k_{0}$	2.884 × 10^−21^	m^2^	$\lambda_{\mathrm{dry}}^{0}$	4.282	W m^−1^ K^−1^
$A_{k}$	0.005	K^−1^	$A_{\lambda }$	−0.002108	K^−1^
*D*	1.319 × 10^−6^	m^2^ s^−1^	$\Delta H_{\mathrm{dehydr}}$	2400	kJ kg^−1^
*a*	52.691	kPa	*h*	238.1	W m^−2^ K^−1^
*b*	1.778	—	ε	1	—
$\beta_{c}$	0.2	m s^−1^	$C_{p0}^{s}$	1200	J kg^−1^ K^−1^

## Data Availability

The most relevant data were provided in the article. All input data, source code, parameters and scripts are available at Zenodo, via the DOI 10.5281/zenodo.4452800 (accessed on 27 August 2021).

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
