# Peer review of "A Three-Phase Transport Model for High-Temperature Concrete Simulations Validated with X-ray CT Data"

_materials, 2021, doi:10.3390/ma14175047_

Round 1

Reviewer 1 Report

This research work shows an analytical study on a three-phase transport modelling for high-temperature concrete, then validated the model using X-ray CT data. Overall the paper is properly written with the only change should be made on the introduction part  My comments are listed as follows:

  1. Although the three-phase model is not new, the non-destructive CT measurement is novel for this type of experiments. However, in the introduction part the literature review of using non-destructive CT to characterize concrete properties under a high temperature is not sufficient.
  2. The authors need to restate the novelty and the research significance of this work
  3. Did the authors code the program by themselves or use a commercial package.
  4. A list of notations is suggested
  5. The model is well presented.

Author Response

Dear Reviewer,

thank you for taking the time to review our paper, and for your constructive criticism.

Although the three-phase model is not new, the non-destructive CT measurement is novel for this type of experiments. However, in the introduction part the
literature review of using non-destructive CT to characterize concrete properties under a high temperature is not sufficient.

We have added the following paragraph to address this:

    Balázs et al. have used X-ray CT to measure the change in density of
    concrete specimen after undergoing a heating cycle [8]. A reduction in
    density by 2% to 4% was found after heating to 500°C, with good
    agreement between CT and conventional methods. The use of CT can resolve
    these changes in space. In separate fire test, the reduction in density
    primarily occurred close to the surface that was subjected to the fire.
    Henry et al. have used X-ray microtomography to characterize the change in
    total pore space and its connectivity after heating to 600°C for an hour
    [9]. The total pore space was found to be roughly 1.5 times larger, while
    the connectivity had increased by 3.5 times. The influence of air and water
    re-curing on the pore space, as well as the pore size distribution was also
    investigated. In contrast to the experiments in [8,9], data from Powierza
    et al. [10] was obtained in-situ during the high temperature exposure.
    This makes it ideal for validating the water content in the pores to a
    numerical model, and is therefore used here.

The authors need to restate the novelty and the research significance of this work.

We made several changes to better highlight the novelty and significance:

    To overcome these experimental difficulties, non-destructive computer
    tomography measurements is increasingly used.  Using these CT results for
    validation of the thermo-hygral transport model is an important improvement
    pursued in this work.

    Additionally, knowing the water content allows for a deeper investigation
    of the influence of different dehydration formulations, several of which
    are compared in sections 2.1 and 3.2. We are unaware of any such comparison
    in the literature.

    The novelty in this work is that the skeleton mass balance is solved along
    with the other balance equations, avoiding this error and resulting in one
    fewer constitutive equation.  This also allows for dehydration descriptions
    that depend on the pore state, e.g. liquid water saturation or pore
    pressure, a significant extension compared to previous models.  These
    modifications allow solving the coupled problem more accurately, while
    introducing the possibility of more complex dehydration models in the
    future.

Did the authors code the program by themselves or use a commercial package?

The program was written by ourselves, but is based on the open-source finite
element library FEniCS (https://fenicsproject.org/). As mentioned in section 4,
all our code is freely available at Zenodo.

A list of notations is suggested.

We have added a list of notations to the beginning of the paper.

Reviewer 2 Report

The paper is interesting and well structured, but some considerations should be made better.

The authors should limit the paper's conclusions to concretes composed of cement and quartz aggregate. I find it misleading not to mention this in the title and other sections of the paper. Therefore, I suggest making it clear that all considerations are made on concretes made with quartzite aggregate. The title must contain the words quartzite concretes, figure 1 should report only the thermograms of quartzite concretes (so that the mass loss of the binding fraction is clearly visible) as well as in the introduction and in the other sections of the text it should be made clear that the subject of this article is the concretes with quartz aggregate.

The authors should also explain how they treat the thermal and hydric expansion/shrinkage of the concretes, considering also the anisotropic features of concrete specimens and aggregate grains (for example, some information on shape, density, porosity, etc. of the aggregate grains might be useful).

Finally, I find that information on the binding fraction of concrete is scarce as well as the mechanical and physical properties of the analysed concrete specimens are missing.

Author Response

Dear Reviewer,

thank you for taking the time to review our paper, and for your constructive criticism.

The authors should limit the paper's conclusions to concretes composed of cement and quartz aggregate. I find it misleading not to mention this in the title and other sections of the paper. Therefore, I suggest making it clear that all considerations are made on concretes made with quartzite aggregate. The title must contain the words quartzite concretes, figure 1 should report only the thermograms of quartzite concretes (so that the mass loss of the binding fraction is clearly visible) as well as in the introduction and in the other sections of the text it should be made clear that the subject of this article is the concretes with quartz aggregate.

We have made added several sentences to make it more explicit that the model
does not allow for chemical changes of the aggregates.

In the abstract:

    "The aggregates are assumed to be chemically inert."

In the introduction:

    "Lastly, the aggregates are assumed to be chemically inert.
    This is valid for quartz or basalt aggregates, or at temperatures below 800K."

In section 2.1, discussing figure 1:

    "Different aggregate types exhibit different temperature stability. When
    using TGA data to model dehydration, the assumption that mass loss equates
    to water release only holds for aggregate types that are chemically stable
    at high temperatures, such as quartzite or basalt. Below 800 K, most
    aggregates are chemically stable."

In section 3.2:

    "Note that while the concrete under investigation has quartzitic
    aggregates, the comparatively low temperatures of the experiment allow for
    experiment and modelling to be repeated for other aggregate types without
    difficulties."

We would like to stress that the behaviour of the aggregates was not the focus of this work. A comparison of different aggregate types at high temperatures shows small differences between granite, basalt and quartz.

"Hager et al. (2016) The influence of aggregate type on the physical and mechanical properties of high-performance concrete subjected to high temperature. Fire and Materials, 40: 668-682"

The authors should also explain how they treat the thermal and hydric expansion/shrinkage of the concretes, considering also the anisotropic features of concrete specimens and aggregate grains (for example, some information on shape, density, porosity, etc. of the aggregate grains might be useful).

Deformation of the concrete is not included in the model. We have added a small
section to make that clearer to the reader:

    "The material is modeled as an isotropic homogeneous continuum. As a
    macroscopic model, it does not resolve the aggregates as discrete particles
    in the finite element mesh. The focus lies on the heat and moisture
    transport phenomena; coupling to mechanical deformation and damage
    evolution are possible extensions in future work."

Finally, I find that information on the binding fraction of concrete is scarce as well as the mechanical and physical properties of the analysed concrete specimens are missing.

We did not perform the experiments ourselves. The mixture information (already
given in Table 1), along with the compressive strength, is unfortunately all
the information the original authors (Powierza et al, reference 10) have
published about the concrete in question. We have added the compressive
strength to the paper:

    "The concrete has a compressive strength of 104 MPa [26]."

Reviewer 3 Report

The study was carried out at a high level, has scientific novelty and great practical significance. An important advantage of this work is the ability to reproduce all the results by other researchers.

There is one small remark on the article.Many quantities are used in the mathematical model, and it is not entirely clear what some of the designations mean. It would be desirable to make an appendix with the notation used in the formulas.

Author Response

Dear reviewer,

thank you for taking the time to review our paper, and for your constructive criticism.

it is not entirely clear what some of the designations mean

We have added a list of notations to the beginning of the paper.

Round 2

Reviewer 1 Report

The authors haver properly addressed my comments. The paper is now recommended as a publication in the journal.

Author Response

Thanks for the review.

Reviewer 2 Report

I think the manuscript still suffers from the presence of several sentences to be improved and I do not recommend publication in the present form. I hope the following detailed comments will serve the authors to improve the text:

The authors did not accept the suggestion to refer only to some aggregates, but in doing so they had to indicate that their work presupposes ‘The aggregates are assumed to be chemically inert.’ – lines 9-10. What are chemically non-reactive aggregates? The authors should explain to which aggregates their assumption refers.

The title of the article still does not exactly match the content of the manuscript: the model validation is for low-temperature concrete by X-ray CT data (320°C). Please change the title appropriately.

At line 64 the authors affirm that: The material is modeled as an isotropic homogeneous continuum. Do the authors believe that a carbonate aggregate can be considered an isotropic homogeneous continuum?

At lines 67-69 the authors repeat that ‘the aggregates are assumed to be chemically inert’. The following sentence is not completely true: a) basalt is a fine-grained extrusive igneous rock formed from the rapid cooling of low-viscosity lava rich in magnesium and iron, which it is not always a non-reactive aggregate; b) at temperatures below 800 K not all minerals present in an aggregate are non-chemically reactive substances.

Figure 1 is not very representative because the recipes are not indicated, and the mass losses relating to the main volatile components present in a concrete are not indicated. The reader cannot see the losses related to the CSH phases, sulphates, hydroxides, etc. Which minerals are responsible for the mass losses in the 800-1000K range? Could not the mass losses be due to the presence of carbonates in the aggregates? If so, what does it matter to indicate different names in the legend (calcareous, siliceous, limestone)? At lines 84-85, the authors affirm that ‘Different aggregate types exhibit different temperature stability. As stated above, the different TGA curves probably depend not on different aggregates, but on different carbonate contents in the aggregates.

At lines 102-104, the sentence is unclear. As reported by [18]: ‘The density for d-dried C-S-H (corrected for CH) has been measured by water pycnometry to be 2.85 Mg/m3 (S. Brunauer, S.A. Greenberg, The hydration of tricalcium silicate and β-dicalcium silicate at room temperature, Proceedings of the Fourth International Symposium on the Chemistry of Cements, National Bureau of Standards, Washington, D.C., 1960.).

At line 167 the authors should at least indicate all the major hydration products of a cement (Calcium Silicate Hydrate, Calcium Hydroxide, Calcium Aluminate Hydrate and Ettringite).

At lines 274-275 the sentence ‘Dry cement mixed with dry aggregates would result in a density of about 2700 kgm3’ may not be true because it depends on the density of the cement and on the density of the aggregate as well as on their proportions in the mixture.

Author Response

Dear Reviewer, thank you for your critical comments. We believe that several misunderstandings originate from the multidisciplinarity between us. Our focus was on model formulation, numerics and validation. In this regard, mineralogy and cement chemistry is addressed only briefly, where it underlies the input data of the multiphase model. We addressed the comments and added a few references. The authors did not accept the suggestion to refer only to some aggregates, but in doing so they had to indicate that their work presupposes ‘The aggregates are assumed to be chemically inert.’ – lines 9-10. What are chemically non-reactive aggregates? The authors should explain to which aggregates their assumption refers. We have removed the sentence from the abstract, and added further clarification in the introduction: "In the model, the aggregates are assumed to be inert, both chemically and physically, i.e. they release no dehydration water. Thermogravimetric analysis showed that basalt, granite and sandstone loose up to 3.1 % of mass when heated to 1000 °C [11], with similar results reported in [12]. Above 600 °C carbonate aggregates (limestone and dolomite) decompose into CaO and CO2 [12], resulting in additional mass loss. Concretes with four different aggregates exposed up to 1000 °C showed similar temperature-dependent tensile and compressive strength decrease [12]. This indicates that cement paste presents the weakest link in concrete. However, different aggregates contribute to microcrack evolution due to the thermal expansion mismatch and this will impact permeability, sorption isotherms and diffusivity." The title of the article still does not exactly match the content of the manuscript: the model validation is for low-temperature concrete by X-ray CT data (320°C). Please change the title appropriately. We've added the temperature information to the title: "A three-phase transport model for high-temperature concrete simulations validated with X-ray CT data up to 320°C" At line 64 the authors affirm that: The material is modeled as an isotropic homogeneous continuum. Do the authors believe that a carbonate aggregate can be considered an isotropic homogeneous continuum? It is true that carbonate genesis induces anisotropy. However, as long as the aggregates are randomly oriented and the domain under investigation is large enough (such that the scale separability condition holds), the resulting concrete can be modeled as isotropic. This is standard practice. Most of the experiments to determine material parameters are done on a macroscopic level, such that the measured quantities already assume homogenized behavior. At lines 67-69 the authors repeat that ‘the aggregates are assumed to be chemically inert’. The following sentence is not completely true: a) basalt is a fine-grained extrusive igneous rock formed from the rapid cooling of low-viscosity lava rich in magnesium and iron, which it is not always a non-reactive aggregate; b) at temperatures below 800 K not all minerals present in an aggregate are non-chemically reactive substances. You are of course right on both counts. However, the proposed model does not take the reactivity of the aggregate into account, and under the conditions we have outlined above, the error is neglible. Under different circumstances, the model needs to be modified, or a different model needs to be used. Figure 1 is not very representative because the recipes are not indicated, and the mass losses relating to the main volatile components present in a concrete are not indicated. The reader cannot see the losses related to the CSH phases, sulphates, hydroxides, etc. Which minerals are responsible for the mass losses in the 800-1000K range? Could not the mass losses be due to the presence of carbonates in the aggregates? If so, what does it matter to indicate different names in the legend (calcareous, siliceous, limestone)? At lines 84-85, the authors affirm that ‘Different aggregate types exhibit different temperature stability. As stated above, the different TGA curves probably depend not on different aggregates, but on different carbonate contents in the aggregates. Figure 1 was only intended to show that different aggregates behave differently, some are more stable than others, and to give a feel for the temperature ranges of these changes. We feel that the information it presented is not essential to the paper, and the behaviour of the different aggregates is not discussed further in the paper. Therefore, we have decided to remove figure 1. At lines 102-104, the sentence is unclear. As reported by [18]: ‘The density for d-dried C-S-H (corrected for CH) has been measured by water pycnometry to be 2.85 Mg/m3 (S. Brunauer, S.A. Greenberg, The hydration of tricalcium silicate and β-dicalcium silicate at room temperature, Proceedings of the Fourth International Symposium on the Chemistry of Cements, National Bureau of Standards, Washington, D.C., 1960.). We have reformulated the sentence in question to make it clearer: "The idea of a constant skeletal density has been used as a C-S-H bulk density description in the whole isotherm range [18]. In such a case, the C-S-H sheet maintains a constant skeletal density of 2850 kg/m3, and packing of the globules provides the correct C-S-H bulk density for different water contents [21]." At line 167 the authors should at least indicate all the major hydration products of a cement (Calcium Silicate Hydrate, Calcium Hydroxide, Calcium Aluminate Hydrate and Ettringite). We've reformulated the sentence to address this: "Hydrated cement paste is mainly composed of C-S-H, together with portlandite, ettringite and other minor phases. We may extend the idea of a constant C-S-H skeletal density to the whole cement paste and concrete, resulting in a constant skeletal density ρ_s." At lines 274-275 the sentence ‘Dry cement mixed with dry aggregates would result in a density of about 2700 kgm3’ may not be true because it depends on the density of the cement and on the density of the aggregate as well as on their proportions in the mixture. That is of course true. We wanted to emphasize that the change in density of
these models is unrealistic, the absolute values being secondary. The number given
corresponds to the mix used in section 3.1
(see table 1; ~28% cement at 3100kg/m3 + 72% aggregates at 2600kg/m3 = 2740kg/m3).
We tried to clarify that in the text: "Dry cement mixed with dry quartzitic aggregates in the same ratio as the concrete in section 3.2 would result in a density of about 2740 kg/m3."